# Application of Vibrational Spectroscopic Techniques in the Study of the Natural Polysaccharides and Their Cross-Linking Process

**DOI:** 10.3390/ijms24032630

**Published:** 2023-01-30

**Authors:** Barbara Gieroba, Grzegorz Kalisz, Mikolaj Krysa, Maryna Khalavka, Agata Przekora

**Affiliations:** 1Independent Unit of Spectroscopy and Chemical Imaging, Medical University of Lublin, Chodźki 4a Street, 20-093 Lublin, Poland; 2Department of Industrial Technology of Drugs, National University of Pharmacy, Pushkins’ka 63 Street, 61002 Kharkiv, Ukraine; 3Independent Unit of Tissue Engineering and Regenerative Medicine, Medical University of Lublin, Chodźki 1 Street, 20-093 Lublin, Poland

**Keywords:** FTIR, Raman spectroscopy, analytical techniques, cross-linking, natural polymers

## Abstract

Polysaccharides are one of the most abundant natural polymers and their molecular structure influences many crucial characteristics—inter alia hydrophobicity, mechanical, and physicochemical properties. Vibrational spectroscopic techniques, such as infrared (IR) and Raman spectroscopies are excellent tools to study their arrangement during polymerization and cross-linking processes. This review paper summarizes the application of the above-mentioned analytical methods to track the structure of natural polysaccharides, such as cellulose, hemicellulose, glucan, starch, chitosan, dextran, and their derivatives, which affects their industrial and medical use.

## 1. Introduction

The word “polymer” has its origin in the Greek language, in which the prefix poly- means “many” and the suffix -mer means “part”. Polymers are large macromolecules consisting of many repeating subunits called monomers, usually linked by covalent chemical bonds [1]. The number of subunits within the polymer, the presence of differing monomers within the same polymer molecule, relative orientation, and the degree to which regularity appears in the order molecule can vary considerably. Formed molecular chains can be linked to each other by covalent bonds to create a three-dimensional network through a process called cross-linking [2]. Polymers occur in a broad variety of forms and may be subsequently classified based on their physical characteristics. Some identifying features include density, hardness, crystalline structure, tensile strength, solubility, formability, machinability, molecular weight, hydrophobicity, surface charge, and thermal properties [3].

Polymers are classified into natural (biopolymers) and synthetic [4]. Human-made synthetic polymers, most of which were developed and mass-produced for about 70 years, include plastics (polyethylene and polystyrene), adhesives (epoxy glue), paints (acrylics), synthetic rubbers (elastomers), and fibers (nylon, polyester) [5,6]. Natural polymers formed by animals, plants, fungi, algae, and bacteria (microorganisms) are nontoxic, biodegradable, and biocompatible. They are mainly divided into three groups:Polypeptides composed of amino acids units, i.e., collagen, fibrin, fibrinogen, actin, myosin, keratin, elastin, silk, and gelatin;Polynucleotides consisting of nucleotides units, such as helical and linear plasmid DNA and RNA;Polysaccharides built of monosaccharide units [7].

Polysaccharide-based polymers are structurally diverse and heterogeneous groups as a multitude of distinct saccharide isomers, frequently with substituents, such as amino acids, O-acyl, or phosphodiester groups, mixed and linked together by utilizing a range of different chemical bonds, which contribute to exceptional biological activities and diversified functional characteristics [8,9]. The reactions leading to hydrocarbon functionalization of polysaccharides include deacetylation, acylation, hydrolysis of main chain, N-phthaloylation, tosylation, reductive alkylation, N-carboxymethylation, O-carboxyalkylation, alkylation, Schiff base formation, silylation, and graft copolymerization. These modifications should help to establish the structure–property relationship necessary to develop specifically desirable functions [10]. The most common natural unmodified polysaccharides include cellulose, hemicellulose, pectin, starch, chitosan, glucan, dextran, carrageenan, xanthan, glycosaminoglycans, alginates, etc. [11].

Taking into account structural criteria, polysaccharides can be classified as:Homoglycans (consisting of one type of monomer unit) and heteroglycans (consisting of two or more types of monomer units);Linear and branched (with various degrees of branching, such as with few and very long branches regularly or irregularly spaced, with short branches forming clusters, or with branch-on-branch structures—“bush-like”);Neutral (noncharged) or charged (cationic or anionic) [12].

Polysaccharides as macromolecules are rather quite heterogeneous. Depending on the structure, these polymers can have individual properties determined by their monosaccharide building blocks; they may be amorphous or even insoluble in water [13]. 

Polysaccharides broadly utilize two types of chemistry for crosslinking: etherification (where an ether bond is formed between polysaccharide -OH groups and carbon of the upcoming molecule) or esterification (where an ester bond is formed between polysaccharide -OH groups and carboxyl groups of upcoming molecules). Etherification reactions support in the introduction of many lipophilic alkyl groups into the chains and, hence, the reduction in the hydrophilic nature as well as the molecular hydrogen bonding [14]. Esterification reactions have also been used to prepare alkyl derivatives of polysaccharides. These consist of reacting either a neutral polysaccharide with an acyl anhydride or an acyl halide or an acid polysaccharide (i.e., hyaluronic acid) with an alkyl halide. In all cases, the reactions are performed in an organic solvent, such as sulfoxide (DMSO) or dimethylformamide (DMF) [15].

This structural diversity leads to the multiplicity of biological functions and provides a basis for physicochemical and rheological properties that can be utilized for diverse commercial applications [16]. They are widely used in various branches of industry, i.e., food processing and packaging (as thickening, stabilizing, gelling, and emulsifying agents), paper, textile, biotechnology (i.e., cell and enzyme immobilization), cosmetology, pharmacy (as adhesives and sophisticated controlled drug delivery systems), and medicine (wound-healing coatings, tissue engineering scaffolds, and immunomodulatory agents) [17]. 

The nonbiodegradable and nonrenewable plastic as an environmental and public health concern has increased the interest in materials based on biopolymers from renewable resources to reduce its worldwide impact. The biopolymers include naturally occurring macromolecules synthesized from monomers, i.e., proteins, cellulose, starch, and others. Naturally derived polysaccharide polymers, which are characterized by monosaccharide units linked with O-glycosidic linkages, are produced by a variety of organisms: higher plants, algae, fungi, microorganisms, and animals [18]. Their size and complexity vary from simple linear chains to complex, branching structures, depending on their function [19]. It affects many physical properties, such as solubility, viscosity, gelling potential, and/or surface properties. As their origins are natural, they often offer very good biocompatibility and bioactivity, which allows applications in different indications in regenerative medicine [20].

There are many methods of spectroscopic analysis for determining the molecular structure of polysaccharides resulting from different conditions of cross-linking. Among them, the most commonly used and most helpful techniques are infrared (IR), Raman, X-ray photoelectron spectroscopy (XPS), photoacoustic spectroscopy (PAS), nuclear magnetic resonance (NMR), and crystallography.

Potential applications of biopolymers include a variety of biomedical approaches, enlisting implants and wound dressings in the form of hydrogels mimicking the mechanical and biological properties of natural tissue; drug delivery; tissue engineering (scaffolding); immunomodulation; and vaccines (polysaccharide capsules of pneumococcal and meningococcal vaccines). Polysaccharide-based materials are biocompatible and biodegradable, making them suitable for use in the human body. They can also be engineered to have specific properties, e.g., increased viscosity or a targeted drug release profile, porosity, or surface-promoting cell adhesion [21]. Adequate design of products based on polysaccharides can ideally possess as many desired properties as possible, which can be achieved by proper fabrication, resulting in different cross-linking, crystallinity, and surface arrangement. Vibrational spectroscopy techniques can provide useful information on chemical structure, which is summarized in Figure 1 to the most significant properties of polymers, summarizing research reviewed in the manuscript. 

In this review, we want to present the most important applications of spectroscopic techniques in the study of polysaccharides and the complex process of polysaccharide cross-linking, with an emphasis on the vibrational spectroscopic techniques used for the investigation. Finally, we will show the current state and future challenges in the field of chemical analytical methods based on spectroscopy to study the molecular organizations of biological macromolecules, in this case, with particular consideration of carbohydrates.

## 2. Fourier-Transform Infrared (FTIR) Spectroscopy

Infrared (IR) radiation, ranging from 780 nm to 1 mm, is a part of the electromagnetic spectrum that is divided into three band sections: near-infrared (NIR: 0.78–2.5 μm), mid-infrared (MIR: 2.5–25 μm), and far-infrared (FIR: 25–1000 μm) [22,23]. FTIR spectroscopy operates in MIR range and is widely used in the pharmaceutical industry, biomedicine, tissue studies, and engineering process [24,25,26]. It is possible to apply it to measure solutions including polymerization reactions, e.g., in microfluidic devices, but also larger model reactors employing a macro-attenuated total reflection Fourier transform infrared (macro-ATR FTIR) setup [27]. A schematic representation of the basic FTIR setups is shown in Figure 2.

Attenuated total reflectance (ATR) is recently one of the most frequently used sampling techniques. ATR is a contact sampling method, which uses the crystal with a high refractive index (IRE—Internal Reflection Element) and excellent IR transmitting properties. Since ATR spectra represent the chemical structure of the surface of the sample, the limitations of this technique are thickness and homogeneity of the sample [26]. ATR utilizes a property of total internal reflection resulting in an evanescent wave. It means that some amount of the light energy leaves the crystal and reaches a small distance (0.1–5 μm) in the depth of the surface in the form of waves [28]. Another popular fast and noncontact technique is optical photothermal infrared spectroscopy (O-PTIR) that overcomes the IR diffraction limit. A tunable, pulsed mid-IR laser concentrates on the surface of the sample and induces photothermal effects, which are then recorded with the use of a visible probe laser. This technique has many advantages, which include sub-micron spatial resolution, lack of dispersive artefacts of ATR, and no need for thin samples [29]. In turn, nano-FTIR (nanoscale FTIR) delivers IR spectroscopy at the spatial resolution comparable to atomic force microscopy (AFM) (10 nm), providing broadband hyperspectral imaging and nanoscale chemical identification by covering the whole mid-IR fingerprint and functional group spectral region [30]. The next technique is diffuse reflectance infrared Fourier transform spectroscopy (DRIFTS), which is used to measure powders and rough surface solids. The light incident on a sample may cause a single reflection from the surface (specular reflectance) or be multiply reflected, giving rise to diffusely scattered light over a broad area that is applied in DRIFTS analyses. Optics in the DRIFTS accessory are designed to refuse the reflected radiation and gather as much of the diffuse reflected light as possible [31].

The assignments to FTIR bands of described polysaccharides are summarized and presented in Table 1, as a baseline to further reported investigations on cross-linking.

### 2.1. Research on Cellulose and Its Modifications and Conjugates

Vibrational spectroscopy of polysaccharides provides basic knowledge on their structure, properties, biological modifications, and interactions with the environment. Covalent cross-linking is the best way to improve the resilience, swelling, solubility, modulus, and tensile strength of cellulosic polymeric materials [42]. Søren Barsberg presented for the first time the way that the main features of cellulose can be predicted and assigned using IR spectra (vibrational properties) combined with density functional theory (DFT) calculations. Cellulose may occur in simple single-chain form and different crystal forms (Iα and Iβ) called allomorphs, both built up from parallel chains in flat-ribbon conformation with alternating glycosyl units locked in opposite orientation by two intramolecular hydrogen linkages. In case of cellulose, IR spectroscopy can supply information on the Iα/Iβ form coefficient, degree of crystallinity and polymerization of the pyranose unit, orientation of the hydroxyl groups, molecular organization, and interaction with water [36]. FTIR measurements performed by Buhus et al. proved cross-linking of polymeric interpenetrated–interconnected network hydrogels based on carboxymethylcellulose (CMC) and gelatin (GEL). Cross-linking occurred with epichlorohydrin in alkaline environments, which were designed for obtaining controlled-release polymer–drug systems. In this study, the differences in the composition of hydrogels were correlated with the parameters of the cross-linking reaction (agent and pH of the environment) [34]. In turn, the effect of sodium trimetaphosphate (STMP) as a cross-linking agent on the physicochemical and antibacterial properties of biopolymer-based composite films consisting of chitosan and methylcellulose (CS and MC) were investigated by Wang et al. The bands at 1203, 1128, 1095, and 880 cm^−1^ recorded in FTIR spectra ascribed to the phosphate groups (P=O, –PO_2_, –PO_3_, and P-O-P, respectively) confirmed the successful cross-linking between CS, MC, and STMP. The cross-linked films, in contrast to non-crosslinked films, were characterized by an excellent antimicrobial activity (~99%), had increased tensile strength, a higher elongation at break, a lower enzymatic degradation, a lower swelling ratio and solubility, and exhibited a satisfactory preservative effect on fresh-cut wax gourd after three days at room temperature [38]. In another study, self-assembled and cross-linked chitosan/cellulose glutaraldehyde composite materials (CGC) were examined with FTIR, XRD, and 13C solid-state NMR. 

Briefly, synthesis of the CGC was carried out with solution of low molecular weight chitosan and glacial acetic acid, to which cellulose was added, followed by pH adjustment and glutaraldehyde addition. As a result, polysaccharides were arranged by an extensive network of intra- and intermolecular hydrogen bonding, which provided ordered and diverse structures. Applied analytical techniques showed interactions between the amine groups of chitosan and glutaraldehyde (formation of imine bonds) and provided evidence of cellulose−chitosan interactions for the composites. Pore structure, surface area, and morphology of the CGC materials, together with uptake studies, confirmed potential utility for applications in water treatment, nanomedicine, and drug delivery [37]. In recent years, hydrogels combined with natural polymers are commonly used for their biocompatibility and ecofriendly nature. Pectin and carboxymethyl cellulose (CMC) hydrogels were produced using polyethylene glycol 400 (PEG-400) and glycerol as a plasticizer, which enriched the physicochemical properties (increasing the flexibility) of resultant hydrogels. FTIR measurements testified the valuable interaction among the CMC and pectin in both glycerol and PEG-400. PEG-400 had more reactive –OH groups than glycerol, which influenced the bond formation and the intensity of the bands and, in consequence, it provided its unique features by ensuring the capability to entrap molecules with both small and large size [43]. Modified cellulose materials with epichlorohydrin (EP) as a cross-linking agent were prepared by Udoetok et al. X-ray diffraction (XRD), FTIR, and NMR spectroscopy indicated that cellulose took a one-chain triclinic unit cell form with gauche-trans (gt) and trans-gauche (tg) conformations of the glucosyl bonds and hydroxymethyl moieties. Cross-linking of cellulose occurred at the amorphous domains, while crystalline domains were preserved. In DRIFT spectra the signatures of cellulose and its cross-linked forms were highly overlapped. The band intensities (C–O–H and C–O–C asymmetric stretching, C–H stretching, and C–H bending) increased and became sharper with the increase in the linker content of the cellulose materials. The IR results provided the support that the –OH moieties of cellulose underwent cross-linking with the epoxide ring of EP. This led to the emergence of “pillaring effects” contributing to greater surface accessibility of polar functional groups (such as hydroxyl ones) and active sites that supported hydration and adsorption of the cross-linked biopolymer [44]. 

To conclude, the most popular cross-linking agents for cellulose are epichlorohydrin, aldehydes, urea derivatives, and multifunctional carboxylic acids (citric acid is the most often utilized). The physicochemical properties of the cross-linked cellulose fibers highly depended on the concentration of cross-linker solution, applied temperature, and curing time [42]. The evidence of the cellulose cross-linking can be easily monitored by FTIR spectra, in which the carbonyl and C−H band intensity increased. Moreover, the abundant reactive hydroxyl groups (–OH groups) on cellulose molecules lead to high water absorbency and generation of the cross-linked structure.

### 2.2. Studies of the Molecular Organization of Other Plant-Derived Polysaccharides

Plant polysaccharides are a frequent subject of structural and spectroscopic research. They are the by-products of photosynthesis within the plants and can be extracted from different parts of the plants, such as leaves, stems, roots, rhizomes, pods, fruits, seeds, cereals, corms, exudates, and others. The important advantages for the uses of plant polysaccharides include easy availability from nature, as plant resources are abundant, with sustainable and low-cost production, biodegradability, biocompatibility, water solubility, and swelling ability [42]. The structure of a set of cell wall polysaccharides from cereal grains exhibiting variation of the degree of substitution was investigated by P. Robert et al. They studied xylo-oligosaccharides comprising xylose monounits or units disubstituted by arabinose residues. Main component analysis of FTIR spectra of model mixtures of arabinoxylans, arabinogalactans, and β-glucans showed that it was feasible to define the relative proportions of the polymers and degree of substitution of arabinoxylan in such complex biological mixtures. For the first time, it was proven that IR technique was proper for the characterization of principal structural features of pure arabinoxylan and their oligosaccharides [33]. Carboxymethyl tamarind kernel polysaccharide cross-linked with calcium chloride (Ca^2+^ cation) by ionic gelation technique was investigated by Dagar et al. The aim of this study was to elevate the efficacy of prepared polymeric matrices and develop its diclofenac sodium-loaded matrix tablets intended for sustained drug delivery applications. FTIR analysis revealed that the band of -OH stretching in polymeric films was shifted from 3415.10 cm^−1^ to 3429.59 cm^−1^ because of interaction of the carboxylic group of polysaccharides with Ca^2+^ ions, which confirmed successful cross-linking. XRD affirmed its amorphous structure with a low degree of crystallinity [45]. A pH-sensitive polysaccharide matrix of linseed (*Linum usitatissimum*), L. hydrogel (LSH) was fabricated using free radical polymerization with the use of potassium persulfate (KPS) as an initiator, N,N’-methylenebisacrylamide (MBA) as a cross-linking agent, and methacrylic acid (MAA) and acrylic acid (AA) as monomers in order to ensure sustained release of ketoprofen. Different formulations of LSH-co-AA and LSH-co-MAA were prepared by changing the concentration of monomers and cross-linker. Formulated matrices were analyzed by FTIR, XRD, and SEM. FTIR spectra showed the grafting of AA and MAA monomers onto LSH due to the lack of their carboxylic acid carbonyls bands at 1759 and 1739 cm^−1^, accordingly, while the presence of 1722 cm^−1^ indicated grafting to the LSH. XRD measurements indicated retention of crystalline nature of ketoprofen in LSH-co-AA and its amorphous dispersion in LSH-co-MAA [46]. The effects of the chemical constitution (acrylic acid/acrylamide), the concentration of cross-linker (glycerol diacrylate), and the kind of initiation (redox and photoinitiation) of the hydrogels composed of polysaccharides isolated from seed of *Persea americana var. Hass* were characterized by FTIR spectroscopy, differential scanning calorimetry (DSC), and scanning electron microscopy (SEM) by Lara-Valencia et al. The increase in the absorbance bands at 1677 and 1572 cm^−1^ detected in FTIR spectra confirmed a successful incorporation of polysaccharides into the polymeric network [47]. However, it is not always possible to analyze by FTIR spectroscopy the structural changes in plant-derived polysaccharides at the molecular level and not all absorption bands allow differentiation. Their complex structures, compositions, glycosidic linkage patterns, and the interactions between polysaccharides (or even with polyphenols and proteins) make the application of ATR-FTIR to study cross-linking of plant polysaccharides still challenging [42].

### 2.3. Polysaccharides Cross-Linking Process in the Context of Cell Scaffold Production and Interactions

Polysaccharides, because of their biocompatibility, biodegradability, hydrophilicity, abundance, and presence of derivatizable functional groups, are excellent scaffold materials. Mechanical properties, which ensure proper cell adhesion, growth, and differentiation, are evaluated in terms of gel strength, porosity, Young modulus, etc. These properties can be modulated either by changing the cross-linking density and molecular weight of the polymer or through incorporation of additional components [42]. The addition of proteins to polysaccharides and their junction in more complex structures, mimicking tissues and extracellular matrix, enables the amelioration of gel property through physical cross-linking of constituents and the alteration of gel network structure. Such polysaccharides that are perfect for combining with proteins are carrageenans and glycosaminoglycans (GAGs). In order to noninvasively study the structure of proteins and carrageenans, the FTIR spectroscopy linked with molecular modeling seems to be the perfect tool, because, in the FTIR spectra, the structural bands of protein–polysaccharide complexation in the cross-linked gel network do not overlap, so that the impact of one component of mixture on the other one is observed in the spectrum as the band’s broadening or shifts. More in-depth discussion can be found in the following review [48]. A green polymer composed of carboxymethyl and carrageenan cross-linked with monochloroacetic acid was investigated with ATR-FTIR spectroscopy and 1H NMR to affirm the substitution of targeted functional group in carrageenan. The three new bands confirmed the successful carboxymethylation of carrageenan: 1595 cm^−1^ and 1418 cm^−1^ related to the asymmetrical and symmetrical stretching vibrations of carboxylate anions (-COO-) and the bands at 1326 cm^−1^ attributed to the –CH_2_ scissoring. Electrochemical impedance spectroscopy (EIS) shows that mixed carboxymethyl and carrageenan film has better ionic conductivity than carrageenan film [35]. Chemically cross-linked three-dimensional (3D) porous scaffolds composed of nano-fibrillated cellulose, carboxymethyl cellulose, and citric acid (CA) produced by combining direct-ink-writing 3D printing, freeze-drying, and dehydrothermal heat-assisted cross-linking techniques were designed by Štiglic et al. Studies have shown that elastic modulus, compressive strength, and shape recovery of the cross-linked scaffolds increased remarkably along with increasing cross-linker concentration (2.5–10.0 wt.% of CA). Surface properties confirmed by SEM images were as follows: 86% porosity and 100-450 µm pores. Furthermore, cross-linked scaffolds favored cell proliferation and exhibited no cytotoxic properties in MTT assay, which proved its biocompatibility [49]. In the study conducted by Hinsenkamp et al., hyaluronic acid (HA) was cross-linked with divinyl sulfone (DVS) and butanediol diglycidyl ether. Fibrin and serum from platelet-rich fibrin (SPRF) were combined with the biocompatible scaffold to promote cell adhesion. FTIR measurements displayed that addition of fibrin into the gel was more efficient than linking SPRF. When fibrin was added, two new intense bands appeared at ~1645 and ~1530 cm^−1^; they were presumably due to the new amine groups of the serum fraction that occurred also in fibrin, as both absorbance wavelengths were specific for fibrin. They should not be confused with the band at 1606 cm^−1^ assigned to the amide group in pure HA. It is also noteworthy that decrease in the band at 1030 cm^−1^ likely means that the fibrin constituted a potent coating. A 5% DVS-crosslinked fibrin-containing hydrogel was the most promising derivative; therefore, it was injected subcutaneously into C57BL/6 mice for 12 weeks, where remodeling and vascularization occurred, and shape and integrity were maintained [50]. UV radiation synthesis and investigations on chitosan-based hydrogels modified with Aloe vera juice were presented by Drabczyk et al. Chemical structure of polymers was studied by FTIR spectroscopy and the spectra revealed that the band of the –OH (and –NH) stretching mode in the 3000–3500 cm^−1^ range was nearly undetectable. This may be due to the fact that mentioned groups form bonds with other functional groups and, hereby, a cross-linked polymer network was created. The choice of the proper conditions for the synthesis of hydrogels allowed polymers with good strength parameters to be obtained with appropriate flexibility and high surface roughness confirmed by texture analyzer and AFM. Furthermore, these materials did not exhibit cytotoxicity towards eukaryotic cells (L929 murine fibroblasts) in MTT assay. Taking into account all these properties, fabricated hydrogels were an attractive material for biomedical applications, i.e., as dressing materials [51]. A series of porous blocks formed of collagen with different concentrations cross-linked with oxidized gum Arabic possessing aldehyde groups was fabricated and studied by Rekulapally et al. The oxidation level of gum Arabic influenced the covalent incorporation of the polysaccharides into the scaffold; this, in turn, resulted in variation in polysaccharide to protein content in scaffolds. The scaffolds were examined for their physical properties, stability, biocompatibility, and ability to support the cell growth. FTIR spectroscopy measurements indicated the formation of the scaffold—the amide I and amide II bands between 1013 cm^−1^ and 1033 cm^−1^ were clearly visible, indicating the presence of C–O stretching (C–O–C). Disappearance of the band in the 1725–1735 cm^−1^ range in the scaffold suggested that the aldehyde group participated in the scaffold formation. The matrices could be designed according to the tissue engineering requirement and stimulate extracellular matrix (ECM) production by cells [52]. 

FTIR spectroscopy is a very versatile technique that provides biochemical information regarding both the molecular arrangement of tissue engineering products as well as the interactions of cell components, such as proteins, lipids, nucleic acids and carbohydrates with biopolymers and cell scaffolds. Furthermore, this approach is highly useful for assessing the ECM surface modifications of biomaterial and the identification of changes resulting from culturing cells on a composite surface [53].

### 2.4. Mechanism of Polymerization of Chitosan and 1,3-β-D-glucan

Chitosan/1,3-β-D-glucan polymeric matrices have been commonly used in various biomedical applications, e.g., for wound healing or as a base for the production of bone implants [54,55,56,57]. Within the series of study designed and performed by Gieroba et al., the structural changes in hybrid polysaccharide chitosan/1,3-β-D-glucan matrices cross-linked at 70 °C, 80 °C [58], and 90 °C were investigated [59]. In order to achieve this purpose, varied spectroscopic and microscopic techniques, such as ATR FTIR, Raman spectroscopy, XPS, and AFM, were utilized. The results showed the involvement mainly of the C-C and C-H groups and C=O⋯HN moieties in the process of biomaterial cross-linking. Moreover, strong chemical interactions and ionocovalent bonds between the N-glucosamine moieties of chitosan and 1,3-β-d-glucan units were demonstrated (the disappearance of a shoulder band in FTIR spectrum at approximately 3610 cm^−1^ found in chitosan). The strength of interactions increased with the rise of temperature. The same analytical approaches by the same authors were used to determine the effect of different gelation temperatures (80 °C and 90 °C) on the structural arrangements in pure 1,3-β-d-glucan (curdlan) matrices. Conducted analyses showed that curdlan gelled at 80 °C possessed different properties than the sample produced at 90 °C. The second-order derivatives from FTIR spectra indicated different secondary structures in these samples [39]. The analysis of curdlan was also performed by Mangolim et al. with complementary FTIR and Raman spectroscopy [60] during gelling of polysaccharide. They indicated, similarly with other literature reports, bands at 1200–890 cm^−1^ characteristic for curdlan and designating strong influence of molecule hydration in this region [61]. Hydrogen bonds and hydrophobic interactions in low-set and high-set gel, however, were not possible to be observed with infrared spectroscopy, thus not delivering information on degree of cross-linking. That is why the authors implemented complementary Raman spectroscopy with the investigated material, described in the following section. The formulation of chitosan (CS) matrices with vanillin as a natural cross-linking agent was optimized in order to improve their physical, barrier, mechanical, and antioxidant properties by Tomadoni et al. Three parameters were analyzed: glycerol content (30, 45, and 60% *w*/*w* chitosan), vanillin content (0, 25, and 50% *w*/*w* chitosan), and drying temperature (35, 50, and 65 °C). FTIR spectra proved that, in the absence of vanillin, there were hydrogen bonds between the -OH moieties of chitosan and glycerol, thereby decreasing the solubility of polymer in the water. An optimal chitosan film formulation was the one with glycerol content of 45% (*w*/*w* of CH), vanillin content of 37.5% (*w*/*w* of CH), and drying temperature of 57.5 °C [62]. In the study conducted by Acharyulu et al., chitosan was cross-linked with glutaraldehyde and blended with the copolymer polyacrylonitrile in different ratios (1:1, 1:2, 1:3, 2:1, and 3:1), and then was characterized by FTIR, DSC, XRD, and thermogravimetric analysis (TGA) techniques. FTIR spectra of unmodified and modified chitosan films (the appearance of the band at 1604 cm^−1^ ascribed to CH_3_(CH_2_)_2_HC=N- stretching vibration) proved the formation of covalent and ionic bonds testifying to cross-linking between chitosan and glutaraldehyde and polyacrylonitrile. Additionally, the created cross-linked chitosan–polyacrylonitrile material was characterized by good thermal stability and satisfactory adsorption capacity [63].

Spectroscopic techniques may be used as inexpensive, fast, and requiring-no-preparation tools to determine the molecular structure of biopolymers consisting of chitosan and 1,3-β-d-glucan, determining whether they have the appropriate parameters for biomedical applications [59].

### 2.5. Molecular Structure of Other Polysaccharides

Polysaccharides are a very complex group and all of them can be studied by spectroscopic techniques in order to determine their cross-linking pattern. FTIR and 13C solid-state nuclear magnetic resonance (NMR) spectroscopies were applied to analyze the alterations in the dextrin structure (single helical and ring conformations) taking place during hydrogel formation by vinyl acrylate (VA) grafting and following free radical polymerization. Both FTIR and NMR techniques demonstrated that, with the increase in %VA, structural changes occurred in dextrin—an ordered single helical conformation prevailed for low %VA polymers and was replaced by a disordered structure as %VA increased. Moreover, analysis of FTIR spectra highlighted that ring conformations were especially influenced by polymerization [64]. In the research conducted by Garcia and Masueli, pectin (Pec) was cross-linked with two different agents: calcium chloride (II) and iron chloride (III). Surface morphology (SEM analysis) and mechanical properties (tensile strength and elastic modulus) were investigated and permeation assays using N_2_, O_2_ and CO_2_ gases were performed. From FTIR analysis, esterified methyl groups and carboxylic acid groups were recorded at 1650 cm^−1^ and 1750 cm^−1^, respectively. Generally, the arrangement of two-folded chains in Pec-Ca with regard to the three-folded chains in Pec-Fe were in charge for acquiring stronger mechanical properties, lower water vapor permeation and water uptake, and favorable O_2_/CO_2_ selectivity in this cross-linked film [65]. The assessment of the influence of fish gelatin–citric acid nucleophilic substitution and agar–citric acid esterification reactions on the physicochemical characteristics of agar/fish gelatin films was performed by Uranga et al. Polymers were gelled at 90 °C and 105 °C and film properties were compared to those of nontreated films. It was stated that temperature fostered the abovementioned reactions, which evoked morphological and physical changes. Therefore, darker films with a rougher surface were obtained for the polymeric matrices with a higher cross-linking degree. The shift in amide II bands in FTIR spectra (from 1549 cm^−1^ for control films to 1537 cm^−1^ for the films treated at 105 °C) and the absence of the two characteristic bands of citric acid at 1690 cm^−1^ and 1743 cm^−1^ attributed to the C=O stretching proved that the temperature promoted cross-linking of the citric acid with the biopolymer. Hence, these agar/fish gelatin films cross-linked via two different reactions can be considered to become promising materials as active polymers for various industrial and medical purposes in the future [66]. In the research conducted by Chen et al., a compound of glycerol (G) and tapioca starch (S) cross-linked with citric acid (CA) at various ratios (0, 0.4, 0.6, 0.8, and 1.0 wt.% of dry basis) was investigated in terms of tensile properties and thermal stability. The appearance of a new band at 1720 cm^−1^ ascribed to the ester C=O stretching in FTIR spectrum upon addition of CA evidenced the esterification of CA, proving that the film was cross-linked. It turned out that specimens with higher CA contents (i.e., 0.8 wt.% and 1.0 wt.%) had undergone the acid hydrolysis on the cross-linking site, reducing the degree of cross-linking [67].

Overall, infrared spectroscopy is primarily applied to evaluate the functional groups of polysaccharides. For instance, bands at 3600−3200 cm^−1^ (ascribed to the angular deformation of the C–H bond), 3000−2800 cm^−1^ (corresponding to C-H stretching vibration in -CH_2_ and -CH_3_ groups, usually present in hexoses, such as glucose or galactose, and deoxyhexoses, such as rhamnose or fucose), and 1200−1000 cm^−1^ (attributed to the stretching vibrations of pyranose ring and C-O-C and C-O-H moieties) are characteristic absorption peaks of polysaccharides. Moreover, a distinct broad peak at 3500−3000 cm^−1^ is ascribed to the O-H bond stretching vibrational mode. Bands in this wavelength can be used to assess whether the polysaccharide is fully methylated [68,69].

## 3. Raman Spectroscopy (RS)

Raman spectroscopy is a powerful and versatile tool for characterizing materials measuring nonelastic light scattering of constantly vibrating intramolecular bonds [70]. Each molecule, after excitation with incident high-power laser light, acquires a “virtual state” and immediately releases energy in different ways, among them, with nonelastic scattering [71]. This light scattered inelastically results in a shift in frequency, which is called Raman scattering and it can be detected and analyzed to provide information about the molecular structure of the sample. In the process, Raman spectra are acquired, including the “molecular fingerprint” of the investigated sample, which can provide valuable information about its chemical composition and structure [72,73]. RS is widely used in biomedicine. For example, it can be used to analyze the composition of cells and tissues, which can provide important information about the health and function of those cells. It can also be used to study the interactions between drugs and biomolecules or to study the structure and conformation of proteins, which is important for understanding their function. Overall, the ability of Raman spectroscopy to provide detailed information about the molecular composition of biological samples makes it a valuable tool in the field of biomedicine. The naturally derived polysaccharide polymers can be evaluated with Raman spectroscopy by identification of functional groups and, therefore, their chemical composition, which enables understanding its chemical and physical properties [59]. There are several techniques used for instrumental analysis of polymers: Raman spectroscopic imaging—microspectroscopy (RSI), surface-enhanced Raman spectroscopy (SERS), coherent anti-Stokes Raman scattering (CARS), and tip-enhanced Raman spectroscopy (TERS). The brief comparison of each Raman technique is presented in Figure 3.

Since the discovery by Nobel laureate Chandrasekhara Venkata Raman and development in the 20th century, Raman spectroscopy has been a reliable nondestructive technique used to analyze materials. The measurements are not affected by the presence of water, which is beneficial compared to infrared spectroscopy. Due to the fact that water is only weakly polarizable and Raman spectroscopy detects the polarizable bonds, it is suitable for studying moisturized samples [74]. Despite the fact that Raman analysis has some difficulties with quantitative robustness and data analysis [75], the qualitative differentiation abilities are very good. It allows collection of chemical, conformational, crystal, and morphological properties of polymer chains [76]. Its nondestructive nature of measurement made it favorable in industry (food and textiles) [77,78], histopathology [79], and biomedical research [80]. Taking into account research focused on unmodified carbohydrate polymers analyzed with Raman spectroscopy, maxima of bands are summarized and presented in Table 2 for better understanding on spectral analyses.

### 3.1. Raman Spectroscopy and Microspectroscopy in the Study of Polysaccharide Cross-Linking

Microspectroscopy is a type of spectroscopy that is used to analyze the chemical composition and physical properties of small samples, typically on the order of micrometers or nanometers in spatial resolution [89]. Acquired spectra are used for compound identification and qualitative evaluation of differences between processed samples. Additionally, Raman spectroscopy allows analysis of modifications, polysaccharide–ligand interactions, polysaccharide conformations, and biomedical applications, such as the polysaccharides that make up the extracellular matrix or bacterial biofilms.

Gelling properties of curdlan derived from *Agrobacterium* were evaluated by Mangolim et al. with Fourier-transformed Raman scattering (FT-Raman) [60]. The aim of the study was to characterize structural arrangement of curdlan alongside degree of polymerization by FTIR and Raman spectroscopy. The authors indicated more clear spectra in the range of 800–1600 cm^−1^ of commercial, bacteria-derived curdlan, both pre-gelled and gelled in 61 and 95 °C. It was discussed that low-set gel had a single-helix structure formed by intramolecular hydrogen bonding of oxygen in glucose ring (C_1_-O-C_5_) and hydroxyl of C_4_ or between hydroxyls of CH_2_ groupings of C_6_ [90]. The high-set gel, however, formed triple-helix conformation, associated to angular deformation of glucose CH_2_ of an intermolecular hydrophobic nature [60]. FT-Raman was concluded to be more sensitive on molecular interactions of low-set and high-set gels, compared to FTIR, referring to fabricated polymer matrices.

As a widely present plant-derived polymer, cellulose and its derivatives were widely investigated using Raman spectroscopy. Most differences in band intensities and band widths were assigned to supramolecular variances: changes in crystallinity, aggregation, and secondary or tertiary structures [91]. There were at least six cellulose polymorphs: cellulose I, II, III_I_, III_II_, IV_I_, and IV_II_ and specific Raman bands associated with cellulose I–III have been reported [92,93]. Schenzel et al. proposed the use of CH_2_ deformation modes (1450–1480cm^−1^) with deconvolution to measure crystallinity of cellulose forms [94]. Highly crystalline and amorphous peaks were assigned at 1481 cm^−1^ and 1462 cm^−1^, respectively, and the intensity ratio (I_c_) was calculated. It was followed with a calibration curve of 0–69% range of crystallinity, allowing experimentally acquired I_c_ to be assigned to crystallinity, even with microcrystals. 

Another plant-derived polysaccharide in the form of thermoplastic starch was meticulously studied by Nobrega et al. [95]. The authors identified the band of polymerization degree at 480 cm^−1^ assigned to the skeletal modes of the pyranose ring [74]. The abovementioned band was one of the dominating in cassava starch Raman spectra. A study by Schuster et al. investigated the Raman signal of starch in response to molecular order loss during gelatinization by FT-Raman [96]. It was indicated that, during gelatinization, water uptake into the crystalline structure of starch was breaking intramolecular hydrogen bonds, forming new ones with water, observable by increasing intensity of bands at 1633 and 3212 cm^−1^. Thus, the crystallinity of starch was able to be evaluated and compared during polymerization. Cross-linking of carboxymethyl starch was evaluated with FT-Raman by Grabowska et al. [97]. Initial observation was the presence of hydrogen bonds in a wide range at 3700–2900cm^−1^ and dehydration in the range 1700–700cm^−1^. Carboxyl groups were identified as a bonding factor by decay of 1338 cm^−1^ in the presence of physical (microwave radiation and thermal holding) and chemical (Ca^2+^ and glutaraldehyde) agents. 

Raman spectroscopy was proven to be useful also in investigation of N-acetylation degree, which is fundamental for chemical and physical properties of polymer [83]. FT-Raman of known deacetylation levels of samples were measured and analyzed in terms of reference point set (at 2885 cm^−1^) and by measuring intensity ratios in unnormalized spectra.

β-glucans are polysaccharides built up of D-glucopyranose residues linked by different types of β-bonds, depending on the source of isolation. Thus, plant-derived β-glucans are a mixture of β-(1,3) and β-(1,4) glycosidic linkage without any branching, while yeast and fungal β-glucans are a linear β-(1,3) residue backbone with long β-(1,6) chains in branches in the case of yeast and with short β-(1,6) branches in the case of fungi. Different β-glucans may have different conformations: random coils, helices, rod-like shapes, worm-like shapes, and aggregates [98]. Raman spectroscopy allows the evaluation of cross-linking in various temperatures (70, 80, and 90 °C), indicating higher density of curdlan fibers in lower temperatures and smoother surface compared to 90 °C [39]. Distinct molecular arrangement and topography of curdlan surface influenced physicochemical properties of polymer product, e.g., thin films. 

Model of three-dimensional polymer matrix in dynamic molecular state with water is a paradigmatic model of polysaccharide hydrogel. Rossi et al. followed this principle to evaluate hydrophilic/hydrophobic chemical groups and the degree of cross-linking for better control of swelling equilibrium using nanosponges of cyclodextrins [99]. UV Raman scattering spectroscopy provided physicochemical bases for design of smart hydrogel carriers for delivery systems. Two intense signals were assigned to ν(C=C)_1_ from ring breathing mode and ν(C=C)_2_ stretching of aromatic ring, but they are practically IR-inactive. 

The approach proposed by Crupi et al. was cyclodextrin polymer evaluation prior to entrapment design inside the matrix. It was observed that C=O stretching was involved in the formation of hydrogen-bonded and nonhydrogen-bonded carbonyl groups (ester and carboxylic) [100]. The amount of C=O stretching bands was proportional to cross-linking degree of polymeric cyclodextrins and can be related to bands at 1015 (C-O) and 2903 cm^−1^ (CH_2_) for FTIR and Raman, respectively. They can be used as an internal reference to evaluate cross-linking of polymeric network. Additionally, low-wavenumber Raman shift of amorphous materials prominent in spectra, so-called boson peak (BP), was observed in the range 1–100 cm^−1^. BP was connected to elastic properties of the material, providing information on structural characterization of material. Similar observations were made by Castiglione et al., indicating an IR-inactive nature of those features [101].

The density of produced thin matrices affects the mechanical, thermal, and rheological properties of the polymer product. A highly cross-linked matrix will be stiffer and stronger but also less flexible and less able to dissipate energy. A lightly cross-linked matrix will be more flexible and able to dissipate energy but also weaker and less stiff [102]. It is possible to evaluate the density of polymer arrangement using confocal Raman spectroscopy investigating the heterogeneity and distribution of electrostatic interactions [103]. Collected data can be translated to potential drug release rates [104], while the arrangement of biomaterial surface determines adhesion and differentiation of cells [54]. Additional insight into chemical, structural, and functional form can be applied to investigate, e.g., microbial biofilm consisting mostly of polysaccharide polymers and nucleic acid in restricting microbial activity [105]. Recent publications indicate the increase in complexity of host–material interactions in tissue engineering and regenerative medicine. It is associated with precise, individualized medicine to maximize biocompatibility and recovery time by implementing a multidisciplinary approach combining material engineering, nanotechnology, cell and molecular therapy, and chemical investigation at the stage of designing material properties [39,51,104,106]. 

Spectroscopy can provide multidimensional information on multiple parameters at once, such as the chemical composition and structural properties of the matrix, which allows for a more comprehensive understanding of the investigated system.

### 3.2. Surface-Oriented Analysis of Carbohydrate Polymers’ Matrices

When applied to polymer matrices, Raman spectroscopy can be used to identify the chemical composition and structure of the polymer on the surface, as well as any impurities or additives that may be present. Additionally, the technique can be used to study the surface morphology of the polymer matrix, such as the roughness, crystallinity, and orientation of the polymer chains [107,108].

Raman spectroscopy is useful when applied to polysaccharide mixtures, which applies to classic Raman spectroscopy and surface-oriented techniques (SERS, SRS, CARS, and TERS). Tip-enhanced Raman spectroscopy (TERS) is a technique that combines the spatial resolution of AFM with the molecular information provided by Raman spectroscopy [109]. In TERS, a sharp metallic tip is used to focus the excitation laser onto the sample surface. The tip enhances the local electric field at its apex, which leads to an increase in the Raman scattering signal with a spatial resolution down to a few nanometers. TERS is particularly useful for studying surfaces and thin films, as well as for imaging heterogeneous samples with complex structures, and it is a technically good tool for analysis of polymers and polymer blends. However, the biopolymers of interest in this review have not been investigated with TERS. Agapov et al. reported possible identification and differentiation of polystyrene isotopes in a blend consisting of hydrogenous and deuterated polystyrene [109]. 

The approach coupling AFM, FTIR, and Raman microscopy was published by Gieroba et al. investigating chitosan/curdlan blends in terms of their cross-linking [58,59]. Raman spectroscopy allowed observation of O-CC, C-O-C, C-C=O, and C-N=C bands that were assigned to chemical interactions between chitosan and 1,3-β-D-glucan, forming interconnections between the –NH group of N-acetylglucosamine and/or glucosamine units of chitosan and probably the –OH group of the glucan units in 1,3-β-D-glucan. Cross-linking was intensified in higher temperatures of gelation, confirming the hybrid nature of resulting matrices.

Coherent anti-Stokes Raman spectroscopy (CARS) is another type of Raman spectroscopy technique that uses two lasers with different frequencies to generate a coherent light source, which can provide enhanced sensitivity and improved spatial resolution. It was developed to overcome limitations of spontaneous Raman spectroscopy [110]. CARS is a coherent nonlinear optical process, where the pump (ωpu), the Stokes (ωS), and the probe (ωpr) transitions generate the anti-Stokes photons (ωaS). When the frequency difference matches the resonance frequency of a vibrational mode of a molecule, the anti-Stokes signal is enhanced. An especially useful feature of CARS is its polarization sensitivity related to incident light in a three-dimensional space and submicrometric resolution [111]. It can be useful for chemical imaging of polymer blends [110], organization of molecules, liquid crystals (proteins and membranes), or heterogenous catalysis [111]. CARS was successfully used in the evaluation of non-polysaccharide-based polymers: polyethylene (PE) with the possibility of spatial distinction of deuterated high-density PE and protonated linear low-density PE [112]. Wang et al. evaluated deformation of fibrin (protein polymer) with BCARS microscopy by monitoring three Raman modes corresponding to CH_3_, amide I and phenylalanine by peak positions and widths [113]. CARS was also successfully employed to analysis of drug release from poly(ethyl-co-vinyl acetate) (PEVA) and poly(butyl methacrylate) (PBMA) blend matrix [114] or water diffusion within polymer matrix with deconvolution of O-H band [115]. Its application in cellulose investigation in a polar alignment of chiral glucose units was reported in the literature [116]. However, the authors discussed that CARS data are scarce compared to, e.g., collagen and need further investigations to conclude comprehensively.

Surface-enhanced Raman scattering (SERS) is a technique used to study the molecular structure of a material by measuring the intensity of scattered light from its surface. This technique is particularly useful for studying surfaces and interfaces of polymers. They are particularly well suited to SERS analysis because they often have a very rough or irregular surface. This allows for a strong SERS effect, making it easier to detect and study the molecular structure of the polymer. When the surface is rough or has many irregularities, the scattered light is enhanced, making it easier to detect [117]. 

Analysis of chitosan polymeric matrices is mostly related to fabrication of substrate connected with gold or silver nanoparticles, mostly as drug delivery systems [118,119,120]. Chitosan containing –NH_2_ group increases interaction with metal nanoparticles, enhancing SERS signal. Chitosan was successfully used as a biopolymeric support for SERS, with indication on 1.5% solution used for preparation.

Ivleva et al. introduced SERS to chemical imaging of biofilms made of cellulose, dextran, xanthan, gellan, and alginic acid [105]. Polysaccharides were evaluated as a major component of exopolysaccharide (EPS) matrix composing bacterial biofilms with both SERS and Raman. It was reported that significant differences between techniques were observed. SERS band near 1380 cm^−1^ assigned to symmetric carboxylate stretching was suggested as an EPS marker. Additionally, imaging was performed and metabolic activity of biofilm was suggested to be derived from I_1383_:I_1280_ ratio.

A similar approach to EPS was performed by Cui et al. As polysaccharides play an important role in bacterial adhesion and retention of water and nutrients, SERS analysis was chosen alongside confocal Raman spectroscopy to measure altering biofilm morphology during biofouling [121]. The increase in band intensities at 1095 and 1126 cm^−1^ and appearance of 494 cm^−1^ during exponential growth of biofilm confirmed EPS’s role in biofilm infrastructure. The abovementioned maxima of bands were considered as a marker of polysaccharide film presence in the industrial cleaning process. 

As an example of β-glucans, SERS analysis and the development of new, potent surface for SERS was presented by Barbosa et.al. Botryospheran, a mixed-linked (1→3)(1→6)-β-glucan produced by the ascomyceteous fungus *Botryosphaeria rhodina* MAMB-05, was investigated due to its unique triple-helix conformation in solution and forming of hydrogel films of good resistance, flexibility, and transparency [81]. A thin film was formed with layer-by-layer technique to be used as a substrate for SERS enhancement. Botryospheran was proven by the authors to interact in the form of polymer film with silver nanoparticles and crystal violet, used as a mode analyte. The preparation of polysaccharide substrate did not involve using harmful reagents and organic solvents, providing good spectral data, thus making it green material amplifying SERS up to 106 orders of magnitude.

Overall, Raman spectroscopy is mainly applied to determine the vibrations of polysaccharide molecules, isomers, and atomic nonpolar bonds [68]. It is stated that the C-O-C vibration including α-D-(1→4) linkages is mainly detected in the 960–920 cm^−1^ region. Other crucial ranges for polysaccharide analysis are 1200–1500 cm^−1^ attributed to CH_2_ and C-OH deformation mode, 1200–950 cm^−1^ characteristic for glycosidic bond type, 950–600 cm^−1^ connected with heterocarbon model, and 600–350 cm^−1^ linked with pyranose ring [19,68,122]. 

SERS is commonly used in a variety of applications, including the study of biological molecules, environmental pollutants, and even in the development of new materials. It is a powerful tool for understanding the structure and behavior of polymers and other complex materials. Surface spectral analysis of polysaccharides can be beneficial in a number of ways. It can help to understand the interactions between polysaccharides and biological systems, such as cell surfaces and proteins. This can aid in the development of new polysaccharide-based biomaterials for use in medical devices and tissue engineering. Additionally, surface analysis can also be used to study the mechanisms of action of polysaccharide-based drugs and to develop new, more effective treatments.

### 3.3. Polysaccharide Evaluation in Relation to Cell Activity

The salient purpose of biomedical-related applications must be associated with cell research, as biocompatibility, biodegradability, and nontoxicity are crucial in regenerative medicine [20,123]. The tests are widely used in material science to evaluate the potential applications of new materials in biomedical fields, such as implants, scaffolds, and drug delivery systems. 

Results acquired by Przekora et al. with spectral analysis supplemented with Raman mapping were introduced to evaluate curdlan degradation [124]. The process was chemically induced with hydrochloric acid and compared to activity of osteoclasts, thus proving depolymerization in both cell and cell-free experiment layouts. The polymer was resolved to maltodextrins, glucose, and disaccharides. Distribution of curdlan and carbohydrate monomers was visualized with mapping, providing images coincident with microscopically observed resorption lacunas by scanning electron microscope and fluorescent actin belt staining. The acidic hydrolysis explained the natural mechanism of polymer degradation by osteoclasts and macrophages, which happens after implantation of curdlan-based composites to the damaged bones. 

Chitin and its derivative, chitosan, as a promising agent in regenerative medicine, drug delivery systems, but also in cancer treatment, are highly abundant polymers in nature. They are widely used in the form of films, nanoparticles, and sponges, relatable to potential application [125]. In the case of chitin differentiation of its allomorphs (alpha-, beta-, and gamma-), despite the spectra similarity, the amide I delivers information on principal differences with a detection limit of milligrams. 

Chitosan/1,3-β-D-glucan blend fabricated at 90 °C was investigated with Raman and FTIR spectroscopy, along with cell cytotoxicity tests. It was discussed with results acquired from similar samples gelled at intermediate temperatures, evaluated with spectroscopy and physical chemistry methods [59]. The biocompatibility test performed in direct contact with human fibroblasts proved non-cytotoxicity of the gels, indicating potential in biomedical applications. Raman spectroscopy provided supporting information on thermally stable, stronger intersheets of produced blended matrix. Additionally, the hybrid nature based in strong covalent, ionic, and noncovalent bonds was proven experimentally. Since it was proven that cells are the most inclined to proliferate on polar, positively charged, and slightly rough surfaces [126], the high-temperature blend of polysaccharides is more promising for applications in biomedicine. 

Concurrent analysis with Raman spectroscopy and cell tests on polysaccharide matrices can provide several benefits over studying these properties separately. Acquired data can be correlated with cell activity and structural properties. Additionally, a better understanding of the mechanism of how cells interact with the matrix is possible, leading to optimization of the matrix properties to promote the growth and activity of the cells. 

## 4. Other Spectroscopic Techniques Used for Polymer Analysis

XPS provides a quantitative surface analysis for all elements except hydrogen and helium in the sample under a high vacuum in the outer 5–10 nm of a surface. It is a very potent measurement technique because, apart from indicating what elements are present and prevalent within polysaccharide molecule, it also shows what other elements they are bonded to [127,128]. For example, XPS enables the determination of carbon in alcohol (C-OH) and acetal (O-C-O) functional groups in polysaccharide and calculates their stechiometric ratio [129]. The photoacoustic approach (PAS) has also been used to quantitatively measure macromolecules, such as polysaccharides. This technique consists in the absorption of electromagnetic radiation by a specimen and integrates the properties of optical spectroscopy and ultrasonic tomography [130]. It is a useful method in the investigation of soluble and insoluble polysaccharides in order to study the presence of conjugated chromophores linked to the polymer chains and the degree of oxidation of the macromolecule [131]. 

Other spectroscopic methods worth mentioning for analyzing the chemical composition of samples, although they are not the subject of this review, are NMR and crystallography. NMR is a spectroscopic technique designed to observe local magnetic fields around nuclei of the atoms [132]. It has been extensively used as an analytical chemistry tool to analyze the molecular structure and conformation of polysaccharides; it can detect the linkage and sequence of sugar residues in examined sample, phase changes, solubility, conformational and configurational alterations, and diffusion potential. NMR has also been applied in quantitative analysis, degradation process studies, polysaccharide mixture interaction analysis, and carbohydrates impurity profiling [133,134]. 

Crystallography is a branch of science that investigates the arrangement and bonding of atoms in crystalline solids and the geometric structure of crystal lattices [135]. In terms of polysaccharides, amorphous matrices, polycrystalline fibers, microscopic single crystals, crystalline powders, and semi-crystalline granules could be analyzed by crystallography [27]. 

## 5. Chemometric Analysis in the Polysaccharide Investigation

Chemometrics is a chemical discipline that contains mathematical and statistical techniques used for the data analysis to provide the chemical information from the sample that would not be accessible with the use of classical methods. In the spectroscopic analysis of the polysaccharides, the main chemometric techniques include principal component analysis (PCA) and regression (mainly partial least-squares regression—PLSR) [136]. 

Principal component analysis is a dimensionality reduction technique that enables dimensionality reduction without losing the variation in the data [137]. In the spectroscopy it is mainly used for the separation of the groups of the samples that differ chemically. When the PCA is performed on the raw spectra, the loadings of the principal components might be analyzed and the main differences that separated one group from another might be identified. PCA was proven to be useful in the spectral analysis of the various polymers. In the cellulose analysis, it was used for the differentiation of the raw naturally colored and white cotton cellulose fibers [138]; moreover, it was used to differentiate the cellulose after the treatment of different temperatures to explore the temperature-dependent changes in the hydrogen bonds of cellulose [139]. The PCA technique was also used for the differentiation of the non-cellulosic polysaccharide components isolated from the plants [140] and for the characterization of β-glucans isolated from the barley and oat [141].

Partial least-squares regression is a multivariate calibration method, which enables establishing a linear regression model, predicting the concentration of the analyte in the sample from the samples spectrum [141]. PLSR is a widely used chemometric technique in the spectroscopic investigation of the polysaccharides. It was used to establish cellulose crystallinity [142], β-glucan and starch concentration [141], the level of deacetylation of chitosan [143], or the composition of the alginate [144].

## 6. Conclusions

Polysaccharides are complex bio-molecules that are structural components (i.e., in plant cell walls and crustaceans’ skeletons) and a key source of energy in animal cells. Because they perform many biologically significant functions, understanding their molecular structure seems to be extremely important. Polymerization and cross-linking processes affect the chemical arrangement of polysaccharides, which, in turn, determines their physicochemical properties. Spectroscopic techniques are an excellent tool for studying their organization and various configurations resulting from polymer cross-linking. Certainly, the advantages of spectroscopic analytical methods include high sensitivity, safety, low operating costs, noninvasiveness, and fast, often automated sample measurement. In addition, they require little or no sample preparation and staining. These techniques are very promising for the future due to the continuous technological advancement, allowing for enhancement of the resolution, simultaneous measurement of multiple samples, and reduction in analysis time. FTIR and Raman spectroscopies provide chemically complementary information and, therefore, measurement of samples by all of these techniques provides highly detailed structural characterization. Chemometrics is also a perfect complement to spectroscopy, which is a multidisciplinary approach referring to the analysis of large, complex chemical data obtained from spectra.

## Figures and Tables

**Figure 1 ijms-24-02630-f001:**
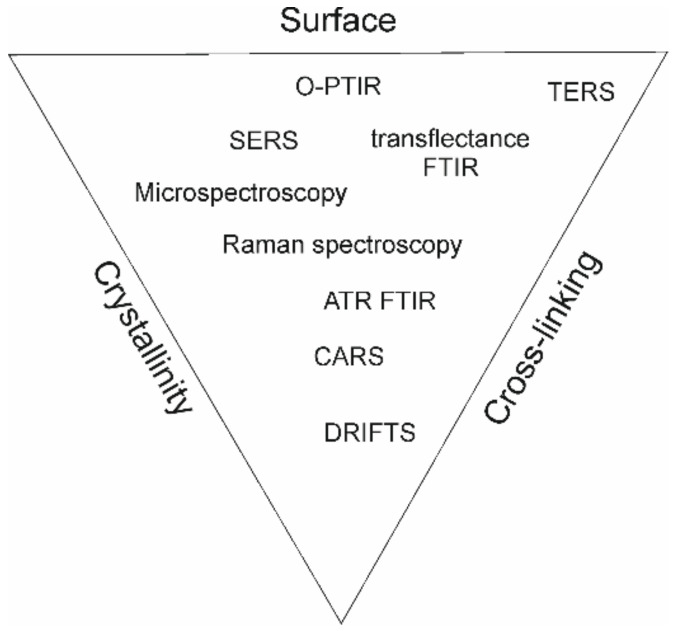
Scheme of polymer properties with relation to spectroscopy techniques reported in the literature.

**Figure 2 ijms-24-02630-f002:**
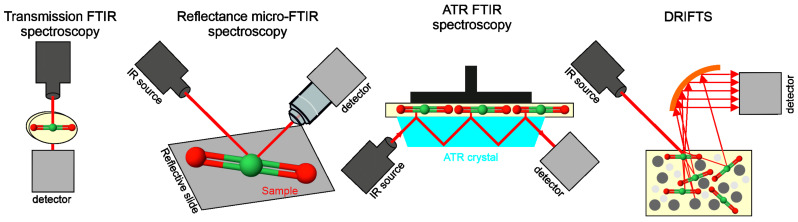
The schematic representations of main FTIR techniques used in analysis of polysaccharide biopolymers.

**Figure 3 ijms-24-02630-f003:**
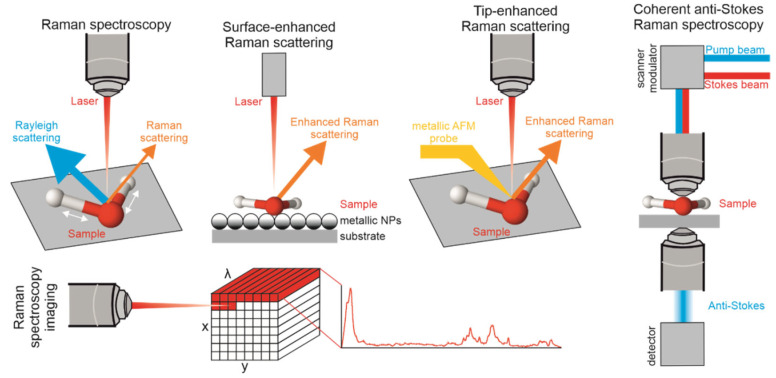
The main Raman techniques implemented to biopolymer analysis: Raman spectroscopy, SERS, TERS, CARS, and Raman spectroscopic imaging (microspectroscopy).

**Table 1 ijms-24-02630-t001:** The most important bands obtained in the FTIR spectra of described polysaccharides.

Wavenumbers (cm^−1^)	Assignment	Wavenumbers (cm^−1^)	Assignment	Wavenumbers (cm^−1^)	Assignment
Agar [32]		Arabinoxylan [33]		Carboxymethylcellulose [34]	
3404	OH stretching	1160	CO asymmetric stretching	3416	OH stretching vibrations
2924	CH_2_ stretching	1045	COH bending	3238	OH stretching vibrations
1392	CH_2_ bending	1020–920	substitution of the xylan backbone by arabinose residues.	2986	CH stretching vibration
1039	CH_2_ scissoring	895	COC stretching (β-1→4 glycosidic bond)	1618	CO asymmetric stretching
				1420	CO symmetric stretching
				1108	COC stretching
				1060	COC stretching
**Carrageenans** [35]		**Cellulose** [36,37]		**Chitosan** [37,38]	
3350	OH stretching	3700–3000	OH stretching vibration	3700–3000	OH stretching vibration
2922	CH stretching	3000–2800	CH stretching	3000–2800	CH stretching
1215	O=S=O symmetric stretching	1427	CH and OH wagging	1632	Amide I (C=O)
1156	CO stretching	1370	C3, C4: CC stretching, CH wagging	1570	Amide II (NH)
1058	CO stretching	1335	CH and OH wagging	1130	Amide III (CN)
928	COC stretching	1314	CH and OH wagging	1080	CO stretching
846	O-SO_3_ stretching	1161	CO asymmetric stretching	1033	CO stretching
		1057	CO stretching		
		1037	C5O stretching		
		983	C1O stretching		
		896	CH rocking		
		610	COC scissoring		
		550	C3,C6: OH torsion		
**Curdlan** [39]		**Hyaluronic acid** [40]		**Pectin** [41]	
2917	CH_2_, CH_3_	3425	OH stretching	3402	OH stretching
2884	CH	2970	CH_3_ asymmetric stretching	2924	CH stretching
1640	C=O	2923	CH_2_ asymmetric stretching	1715	C=O stretching (ester groups)
1576	COO^−^	2880	CH_3_ symmetric stretching	1620	C=O stretching (carboxyl groups)
1463	CH_2_	2853	CH_2_ symmetric stretching	1148–1041	Glycoside bond and sugar ring
1419	CH_2_, CH_3_	1740	C=O stretching		
1367	CH, CH_3_	1657	Amide I		
1312	CH_2_	1555	Amide II		
1285	C-O	1468 and 1454	CH_2_ and CH_3_ scissoring		
1235	C-OH	1427	COH scissoring		
1203	C-O, C-O-C	1380	CH_3_ and OH scissoring		
1157	C-O-C (ring)	1325, 1260, 1229	Amide III		
1107	C-O	1156, 1126	COC stretching		
1067	C-O	1105, 1079	CO and CC stretching, COH scissoring		
1028	C-O	946	CC out of plane vibration		
991	C-O, C-C	925	COH out of plane vibration		
926	C-H	692	CH_2_ out of plane vibration		
887	COC, (β glycosidic bond)				

**Table 2 ijms-24-02630-t002:** The most important bands obtained in the Raman spectra of described polysaccharides.

Wavenumbers (cm^−1^)	Assignment	Wavenumbers (cm^−1^)	Assignment	Wavenumbers (cm^−1^)	Assignment
Botryospheran [81]		Cellulose [82]		Chitosan [83]	
1382	CH_2_ bending OH bending	2968, 2944	CH_2_ asymmetric stretching	2932	CH_3_ stretching
1115	C-O-C antisymmetric stretching	2904–2894	CH stretching	2885	CH_2_ stretching
1100	C-O-C symmetric stretching	1470	CH_2_ bending	1458	CH and OH scissoring, CH_2_ wagging
1067	C-C, C-O stretching	1378–1338	C–C-H, C-O–H, and O-C-H	1411	CH_3_ and CH scissoring
826	stretching of ring carbon atoms and OH	1315	C–C-H, C-O–H, and CH_2_ bending	1377	CH_2_, CH and OH scissoring
		1160	Skeletal deformation	1325	CN stretching and CH scissoring
		1120–1090	Skeletal deformation	1146, 1114, 1093	COC, COH, CCH_2,_ ring stretching, CH scissoring, CH_2_ and CH_3_ rocking
		1050–970	C–C and C-O	1044	CH_3_ rocking, CH and OH scissoring
		896	Glucose ring deformation	936	CN stretching
		740	Glucose ring deformation with bending of glycosidic bond, Iα type cellulose	896	Ring stretching, CH_2_ rocking
		710	Glucose ring deformation with bending of glycosidic bond, Iβ type cellulose	493	CONH and CCH_3_ scissoring,
		380	Out of plane breathing of glucose ring	479	COC scissoring
		2968, 2944	CH_2_ asymmetric stretching	444, 424, 357	OH and ring out-of-plane vibration
		2904–2894	CH stretching	285	CNHC scissoring, OH out-of-plane vibration
		1470	CH_2_ bending		
		1378–1338	C–C-H, C-O–H, and O-C-H		
		1315	C–C-H, C-O–H, and CH_2_ bending		
		1160	Skeletal deformation		
		1120–1090	Skeletal deformation		
		1050–970	C–C and C-O		
		896	Glucose ring deformation		
		740	Glucose ring deformation with bending of glycosidic bond, Iα type cellulose		
		710	Glucose ring deformation with bending of glycosidic bond, Iβ type cellulose		
		380	Out of plane breathing of glucose ring		
**Curdlan** [60]		**Cyclodextrin** [84]		**Dextran** [85]	
1459	CH_2_ deformation	1455	OCH and HCH bending	1348	COH
1369	CH and COH deformation	1415	OCH and CCH bending	1274	Glycosidic bond
1263	CH_2_OH	1375	CCH, OCH and COH bending	1084	COH
1205	CCH	1350	CCH, OCH and COH bending	1024	COH
1117	COC asymmetric stretching	1340	CCH, OCH HCH bending	913	COH
1097	COC symmetric stretching	1250	OCH, COH, CCH bending		
1037	CC and COH stretching	1205	CO stretching and CCH, COH bending		
890	CH deformation (glycosidic bond)	1160	CO and CC stretching, COH bending		
		1080	CO and CC stretching, COH bending		
		1050	CO and CC stretching		
		1010	CC stretching, OCH, CCH, CCO bending		
		950	Skeletal vibration involving α-1,4 linkage		
		850	CCH bending, CO and CC stretching		
		580	Skeletal vibrations		
		480	Skeletal vibrations		
**Gellan** [86]		**Starch** [87]		**Xanthan** [88]	
3065	OH stretching	2973	CH_2_ asymmetric stretching		
2937	CH_3_ stretching	2914	CH_2_ asymmetric stretching	2944	CH stretching
1683	C=C stretching	1463	CH_2_ twisting, CH bending	2907	CH stretching
1628, 1578	ring carboxyl groups stretching	1401	OCH and CCH scissoring	1469	CH bending, wagging
1487, 1419	CH in plane vibration	1383	CH scissoring and bending	1413	CH bending, wagging
1390	ring carboxyl groups stretching	1340	CH_2_ scissoring, COH bending	1369	CO stretching
1174	CH in plane vibration CH_3_ rocking	1261	CH_2_ scissoring, CH_2_OH (side chain) related mod	1276	COH stretching, bending, deformation
1004	CC stretching, CH_3_ rocking	1150	COC stretching (α-1,4-glycosidic linkage)	906	COH stretching
960	ring carboxyl groups stretching and bending	1128	COH and CO stretching, COH scissoring	887	β-glycosidic linkage
856, 820	CC stretching	1087	COC stretching (ring) COH bending		
793	CH out-of-plane vibrations	1056	COH stretching		
742	ring carboxyl groups stretching and bending	941	COC stretching (α-1,4-glycosidic linkage)		
643	ring carboxyl groups bending	911	α-configuration COH bending		
601	COC bending, CH out-of-plane vibrations	862	COC stretching (ring), C1H bending α-configuration		
524	ring carboxyl groups stretching	769	CO stretching		
491	COC bending, in-plane CO vibrations	578	skeletal modes		
410	COO wagging	480	skeletal mode involving COC ring mode, CCO scissoring		
286	out-of-plane CO vibrations	440	CCC scissoring		
233	CH_3_ torsion	411	CCO scissoring		
206	CH_3_ torsion, in-plane CO vibrations				

## Data Availability

Not applicable.

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
