# Peer review of "Application of Vibrational Spectroscopic Techniques in the Study of the Natural Polysaccharides and Their Cross-Linking Process"

_ijms, 2023, doi:10.3390/ijms24032630_

Round 1

Reviewer 1 Report

In the review “Application of vibrational spectroscopic techniques in the study of the natural polysaccharides and their cross-linking process”, a huge amount of work has been done to collect the available literature regarding the application of vibrational methods, mainly IR and Raman spectroscopy, in the analysis of both polysaccharides and their cross-linking products, which have useful properties that could be used in the future in various fields of industry and medicine, therefore, in my opinion , the work is relevant. However, before it will be published, some corrections need to be made.

Sections 2.1, 3.1, 3.2 and 3.3 are more of a listing of the work that has been done previously. However, this is a review and it should have its own analysis of the authors of the article of the literature presented, some kind of concept should be formulated, their own reasoning on this matter should be expressed. Therefore, in my opinion, they should be rewritten with this in mind.

In sections 2.1 and 3.1, the titles only include the use of spectroscopy to analyze polysaccharides. And in the sections themselves, it is more about the use of spectroscopy for the analysis of cross-linking products based on polysaccharides. Therefore, it would be logical to reflect this moment in the titles of the sections.

Section 2.1. is quite large, capacious and contains a lot of useful information, but quite difficult to understand. I think it makes sense to systematize the information of the section in a table in order to facilitate the perception of the material by the reader.

Line 14: Please remove “in nature”. Authors already write that polysaccharides are natural polymers and this is enough.

Line 25: “Part” is in the singular, because -mer is also in the singular.

Line 25-27: It is not entirely clear what the authors meant by the words chains or rings. Rewrite the sentence, please as: “Polymers are large macromolecules consisting of many repeating subunits called monomers, usually linked by covalent chemical bonds."

Line 77-78: It is not entirely clear how the functions performed affect the structure and size of molecules. As a rule, the structure and size determine the functions. Is not it so?

Line 225: It's about dextrin or dextran. Because the next sentence says dextran.

Line 236: “and 1650” is redundant. Delete, please.

Line 236: Please, change “accordingly” to “respectively”.

Line 242: The authors introduce the abbreviations IR and NMR, etc., but there is no decoding for the TGA method. For people who are not great experts in this field, it may not be clear what it is. Therefore, I propose to decipher.

Line 244: Please, check the correct spelling of the formula. If write the formula as СН2-СН2-СН2-НС=N-, then the terminal carbon lacks one valency.

Line 279: “Attenuated Total Reflection Fourier Transform infrared”. This abbreviation appears in line 101, but is not deciphered, and ATR stands for line 106. It is also written here in full and abbreviated. Please, decipher the abbreviation at the first mention, and then use the abbreviation, otherwise why enter it.

Line 280: The same with NMR. For the first time, the abbreviation and its decoding were introduced in line 86. In subsequent mentions, there is no need for a full name.

Line 284-286: After the word "scissoring" most likely there should be a point and then the beginning of a new sentence. And this new sentence would be better rewritten as: “Electrochemical impedance spectroscopy shows that mixed carboxymethyl and carrageenan film has better ionic conductivity than carrageenan film”. This will make it easier to understand.

Line 293: There is already a decoding of the abbreviation SEM in line 185.

Line 304: Please, change “decreased band” to “the decrease in the band”.

Line 316: “Atomic Force Microscopy”. There is a decoding of this method in lines 316, 337, 499, but not in line 120. You can give a decoding of the method only in line 120, and then use abbreviations.

Line 330: “ECM production”. What does this mean?

Line 336-337: “X-ray photoelectron spectroscopy (XPS) and atomic force microscopy (AFM)”. Abbreviation of these methods introduced earlier.

Line 349-351: It is not entirely clear what the authors meant.

Line 356-358: It is not clear how the first two sentences are related. What do the authors mean by glycosidic bond configuration: alpha and beta? Then why are absorption bands given in the next sentence, which are not clear what they refer to in the polysaccharide?

Line 405-411: How are glucose units different from D-glucopyranose residues? Isn't it synonymous? I suggest rewriting: "β-glucans are polysaccharides built up D-glucopyranose residues linked by different types of β-bonds, depending on the source of isolation.”

Originally Fibers 2020, 8, 1; doi:10.3390/fib8010001 is written quite differently regarding the structure of β-glucans from plants, fungal and yeast. I suggest rewriting: “Thus, plant-derived β-glucans are a mixture of β-(1,3) and β-(1,4) glycosidic linkage without any branching, while yeast and fungal β-glucans are a  linear β-(1,3) residues backbone with long  β-(1,6 ) chains in branches in the case of yeast and with short  β-(1,6) branches in the case of fungi. Different β-glucans may have different conformations: random coils, helices, rod-like shapes, worm-like shapes, and aggregates."

Line 423-424: What is this about?

Line 583-584: It is not entirely clear what the authors meant. How are these polysaccharides with non-polar bonds?

Author Response

Reviewer 1

In the review “Application of vibrational spectroscopic techniques in the study of the natural polysaccharides and their cross-linking process”, a huge amount of work has been done to collect the available literature regarding the application of vibrational methods, mainly IR and Raman spectroscopy, in the analysis of both polysaccharides and their cross-linking products, which have useful properties that could be used in the future in various fields of industry and medicine, therefore, in my opinion , the work is relevant. However, before it will be published, some corrections need to be made.

Thank you for appreciating our work. We have made every effort to ensure that our manuscript is valuable and meets the requirements of a comprehensive review article. Thank you very much for your attentive and insightful reading of this manuscript and many valuable comments that have improved the quality of our work. We hope that we will address your concerns to a satisfactory degree.

Sections 2.1, 3.1, 3.2 and 3.3 are more of a listing of the work that has been done previously. However, this is a review and it should have its own analysis of the authors of the article of the literature presented, some kind of concept should be formulated, their own reasoning on this matter should be expressed. Therefore, in my opinion, they should be rewritten with this in mind.

Thank You for this general comment, following it we rewrote the manuscript adding insights on the usefulness of vibrational spectroscopy in terms of cross-linking evaluation, referring to properties of polysaccharide matrices/materials in biomedicine. Thus, paragraph on this topic was added in introduction and accordingly more emphasis was put on it in the text. It all referred to the information on techniques application, as presented in Figure 3 which is now moved to the introduction as well, now we hope the main concept is more clear and highlighted in the manuscript. It needed a lot of manuscript to be rewritten so the changes are difficult to indicate.

In sections 2.1 and 3.1, the titles only include the use of spectroscopy to analyze polysaccharides. And in the sections themselves, it is more about the use of spectroscopy for the analysis of cross-linking products based on polysaccharides. Therefore, it would be logical to reflect this moment in the titles of the sections.

Information on cross-linking was added to the sections titles

Section 2.1. is quite large, capacious and contains a lot of useful information, but quite difficult to understand. I think it makes sense to systematize the information of the section in a table in order to facilitate the perception of the material by the reader.

Thank you for appreciating the substantive value of this part of the manuscript. In order to increase the clarity and readability of 2.1. section, 5 subsections have been separated. In addition, the entire text has been reorganized and some fragments have been moved to another section. In our opinion, due to the variety of presented research, the use of various additional techniques and the study of different cross-linking parameters, and the diverse application of polymers, significantly hinder the preparation of a table that will be reliable and accessible for readers. Instead, we have prepared a table for vibrational spectroscopy related to the specific peak characterization for polysaccharides to increase the value of our review article.

Line 14: Please remove “in nature”. Authors already write that polysaccharides are natural polymers and this is enough.

“in nature” was removed from the sentence.

Line 25: “Part” is in the singular, because -mer is also in the singular.

This is an accurate remark, correction was done.

Line 25-27: It is not entirely clear what the authors meant by the words chains or rings. Rewrite the sentence, please as: “Polymers are large macromolecules consisting of many repeating subunits called monomers, usually linked by covalent chemical bonds."

Parts of the sentence were removed to be more clear.

Line 77-78: It is not entirely clear how the functions performed affect the structure and size of molecules. As a rule, the structure and size determine the functions. Is not it so?

Line 225: It's about dextrin or dextran. Because the next sentence says dextran.

“dextran” was a spelling mistake, it was clearly about dextrin, sorry for the inconvenience.

Line 236: “and 1650” is redundant. Delete, please.

The redundant part was removed from the sentence.

Line 236: Please, change “accordingly” to “respectively”.

The replacement was done.

Line 242: The authors introduce the abbreviations IR and NMR, etc., but there is no decoding for the TGA method. For people who are not great experts in this field, it may not be clear what it is. Therefore, I propose to decipher.

Thermogravimetric analysis (TGA) was added to line 243: “…and then was characterized by FTIR, DSC, XRD and Thermogravimetric analysis (TGA) techniques.”

Line 244: Please, check the correct spelling of the formula. If write the formula as СН2-СН2-СН2-НС=N-, then the terminal carbon lacks one valency.

Thank you for noticing the mistake. It was corrected to CH3(CH2)2HC=N-

Line 279: “Attenuated Total Reflection Fourier Transform infrared”. This abbreviation appears in line 101, but is not deciphered, and ATR stands for line 106. It is also written here in full and abbreviated. Please, decipher the abbreviation at the first mention, and then use the abbreviation, otherwise why enter it.

The deciphering was provided in the first mention and removed from line 282.

Line 280: The same with NMR. For the first time, the abbreviation and its decoding were introduced in line 86. In subsequent mentions, there is no need for a full name.

Change in a similar manner as above was made for NMR in line 282.

Line 284-286: After the word "scissoring" most likely there should be a point and then the beginning of a new sentence. And this new sentence would be better rewritten as: “Electrochemical impedance spectroscopy shows that mixed carboxymethyl and carrageenan film has better ionic conductivity than carrageenan film”. This will make it easier to understand.

The sentence was corrected according to suggestion.

Line 293: There is already a decoding of the abbreviation SEM in line 185.

“Scanning electron microscopy” was removed from the sentence.

Line 304: Please, change “decreased band” to “the decrease in the band”.

The sentence was corrected according to suggestion.

Line 316: “Atomic Force Microscopy”. There is a decoding of this method in lines 316, 337, 499, but not in line 120. You can give a decoding of the method only in line 120, and then use abbreviations.

The deciphering was provided in line 120 and only abbreviations were used in the following manuscript.

Line 330: “ECM production”. What does this mean?

ECM is the abbreviation for extracellular matrix, the deciphering was provided in the sentence.

Line 336-337: “X-ray photoelectron spectroscopy (XPS) and atomic force microscopy (AFM)”. Abbreviation of these methods introduced earlier.

The correction was done by removal of X-ray photoelectron spectroscopy and atomic force microscopy.

Line 349-351: It is not entirely clear what the authors meant.

The sentence has been reformulated and now sounds clearer: “They indicated, similarly with other literature reports, bands at 1200-890 cm-1 characteristic for curdlan and designating strong influence of molecule hydration in this region”.

Line 356-358: It is not clear how the first two sentences are related. What do the authors mean by glycosidic bond configuration: alpha and beta? Then why are absorption bands given in the next sentence, which are not clear what they refer to in the polysaccharide?

This information referred to functional groups, not glycosidic bonds. The mistake was corrected:

“Overall, infrared spectroscopy is primarily applied to evaluate the functional groups of polysaccharides. For instance, bands at 3600−3200 cm−1 (ascribed to the angular deformation of the C–H bond), 3000−2800 cm−1 (corresponding to C-H stretching vibration in -CH2 and -CH3 groups, usually present in hexoses, like glucose or galactose, and deoxyhexoses like rhamnose or fucose), and 1200−1000 cm−1 (attributed to the stretching vibrations of pyranose ring and C-O-C and C-O-H moieties) are characteristic absorption peaks of polysaccharides.”

Line 405-411: How are glucose units different from D-glucopyranose residues? Isn't it synonymous? I suggest rewriting: "β-glucans are polysaccharides built up D-glucopyranose residues linked by different types of β-bonds, depending on the source of isolation.”

You are right, glucose units and D-glucopyranose units are synonymous. The correction was made in the manuscript.

Originally Fibers 2020, 8, 1; doi:10.3390/fib8010001 is written quite differently regarding the structure of β-glucans from plants, fungal and yeast. I suggest rewriting: “Thus, plant-derived β-glucans are a mixture of β-(1,3) and β-(1,4) glycosidic linkage without any branching, while yeast and fungal β-glucans are a  linear β-(1,3) residues backbone with long  β-(1,6 ) chains in branches in the case of yeast and with short  β-(1,6) branches in the case of fungi. Different β-glucans may have different conformations: random coils, helices, rod-like shapes, worm-like shapes, and aggregates."

Thank you for a really good point. It was rewritten as suggested.

Line 423-424: What is this about?

This sentence concerns evaluation of gelling properties of curdlan derived from Agrobacterium (study conducted by Mangolim et al.). The phrase “food product” was replaced by “curdlan”.

Line 583-584: It is not entirely clear what the authors meant. How are these polysaccharides with non-polar bonds?

It was corrected. Now the sentence is: “Overall, Raman spectroscopy is mainly applied to determine the vibrations of polysaccharide molecules, isomers, and atomic non-polar bonds”.

Reviewer 2 Report

The paper describes the spectroscopical methods for elucidating the structural features of polysaccharides.

Structure of the paper is written extensively and encompasses various structural characterization.

There are some suggestions for authors that would improve the readability of paper.

1. Please provide tables for vibrational spectroscopy related to the specific peak characterization for polysaccharides. As, such spectroscopy evaluations are commonly employed by the researchers and therefore it would help those researchers to find such information in the paper and that would enhance the readability of the paper.

2. Authors talked about cellulose-chitosan conjugate, but no structural characterization was reported. Please provide sufficient information. Also, such cross-linking or conjugation is challenging therefore provide at least the names of reagents/chemical methods (as briefly as it can be possible) used to achieve such processes.

3. Polysaccharides broadly utilize two types of chemistry for crosslinking: etherification (where ether bond forms between polysaccharide-OH groups and carbon of upcoming molecule) or esterification (where ester bond forms between polysaccharide-OH groups and carboxyl groups of upcoming molecules). Authors needs to add such information to the paper.

4. Author needs to write more about the hydrocarbon functionalization to the polysaccharide, which is inferiorly written in the current form.

Author Response

Reviewer 2
The paper describes the spectroscopical methods for elucidating the structural features of polysaccharides.
Structure of the paper is written extensively and encompasses various structural characterization.
Thank you for taking the time to read our research profoundly. We greatly appreciate such positive feedback about our work.
There are some suggestions for authors that would improve the readability of paper.
1. Please provide tables for vibrational spectroscopy related to the specific peak characterization for polysaccharides. As, such spectroscopy evaluations are commonly employed by the researchers and therefore it would help those researchers to find such information in the paper and that would enhance the readability of the paper.
On basis of analyzed literature reports two tables was added to FTIR and Raman spectroscopy paragraphs, indicating referenced data. We hope this enhances readability of the paper significantly
2. Authors talked about cellulose-chitosan conjugate, but no structural characterization was reported. Please provide sufficient information. Also, such cross-linking or conjugation is challenging therefore provide at least the names of reagents/chemical methods (as briefly as it can be possible) used to achieve such processes.
Both chemical structure as well as fabrication methods were concisely presented as described below:
The fabrication was done as follows: Briefly, synthesis of the CGC was carried out by dissolving  low molecular weight chitosan with stirring in glacial acetic acid using a beaker. Cellulose  was added to the resulting light yellow chitosan solution with continuous stirring. The mixture was raised to pH 5.60 using NaOH before dropwise addition of glutaraldehyde. Then, NaOH solution was added gradually to the mixture with vigorous magnetic stirring until pH 7. The chemical structure of obtained chitosan/cellulose composite is presented in Figure X below, as provided in paper by Udoetok et al : 

Figure X. The structural scheme of chitosan/cellulose conjugate. The purple color corresponds to cellulose and green corresponds to chitosan. 
These polysaccharides are interconnected by extensive network of intra- and intermolecular hydrogen bonding which provide ordered and diverse structures.”
We added paragraph in the manuscript: Briefly, synthesis of the CGC was carried out with solution of low-molecular weight chitosan and glacial acetic acid, to which cellulose was added followed by pH adjustment and glutaraldehyde addition. As a result, polysaccharides were arranged by an extensive network of intra- and intermolecular hydrogen bonding which provided ordered and diverse structures. Applied analytical… in lines 202-206.

3. Polysaccharides broadly utilize two types of chemistry for crosslinking: etherification (where ether bond forms between polysaccharide-OH groups and carbon of upcoming molecule) or esterification (where ester bond forms between polysaccharide-OH groups and carboxyl groups of upcoming molecules). Authors needs to add such information to the paper.
Paragraph concerning etherification and esterification was added in lines 70-80.
“Polysaccharides broadly utilize two types of chemistry for crosslinking: etherification (where ether bond forms between polysaccharide -OH groups and carbon of up-coming molecule) or esterification (where ester bond forms between polysaccharide-OH groups and carboxyl groups of upcoming molecules). Etherification reactions support in the introduction of many lipophilic alkyl groups into the chains and hence the reduction of the hydrophilic nature as well as the molecular hydrogen bonding. Esterification reactions have also been used to prepare alkyl derivatives of polysaccharides. These consist in reacting either a neutral polysaccharide with an acyl anhydride or an acyl halide or, an acid polysaccharide (i.e. hyaluronic acid) with an alkyl halide. In all cases, the reactions are performed in an organic solvent such as sulfoxide (DMSO) or dimethylformamide (DMF).”
4. Author needs to write more about the hydrocarbon functionalization to the polysaccharide, which is inferiorly written in the current form.
This information was extended and written in the lines 52-58.
“The reactions leading to hydrocarbon functionalization of polysaccharides include deacetylation, acylation, hydrolysis of main chain, N-phthaloylation, tosylation, reductive alkylation, N-carboxymethylation, O-carboxyalkylation, alkylation, Schiff base formation, silylation, and graft copolymerization. These modifications should help to establish the structure-property relationship necessary to develop specifically desirable functions”

Round 2

Reviewer 1 Report

The authors have done a great job of revising the article, according to the comments, and significantly improved it, making it more structured and easier to understand. Therefore, I believe that the article can be accepted for publication after minor comments.

Section 3.1., Page 12, paragraph beginning with "Another plant-derived polysaccharide….." For some reason, the same thought is repeated twice. Check, please.

Author Response

Rewiever 1

The authors have done a great job of revising the article, according to the comments, and significantly improved it, making it more structured and easier to understand. Therefore, I believe that the article can be accepted for publication after minor comments.

Thank you for taking the time to read our revised manuscript profoundly. We have put a lot of work into improving it. We greatly appreciate such positive feedback about our research.

Section 3.1., Page 12, paragraph beginning with "Another plant-derived polysaccharide….." For some reason, the same thought is repeated twice. Check, please.

Thank you for noticing that. We have checked Section 3.1. The same thought was written twice by mistake. We have removed one repetition.
